# External cephalic version at 38 weeks' gestation at a specialized German single center

**Ann-Sophie Zielbauer** *, **Frank Louwen, Lukas Jennewein**

Department of Gynecology and Obstetrics, School of Medicine, Goethe-University, Frankfurt, Germany

* ann-sophie.zielbauer@kgu.de

## Abstract

### Introduction

Cesarean section (CS) rates are increasing worldwide. One constant indication is the breech presentation at term. By offering external cephalic version (ECV) and vaginal breech delivery CS rates can be further reduced.

### Objective

This study aimed to analyze the ECV at 38 weeks of gestation with the associate uptake rate, predicting factors, success rate, and complications at a tertiary healthcare provider in Germany specializing in vaginal breech delivery.

### Methods

We conducted a prospective cohort study with retrospective data acquisition. All women with a singleton fetus in breech presentation presenting after 34 weeks of gestation for counseling between 2013 and 2017 were included. ECV impact factors were analyzed using logistic regression.

### Results

A total of 1,598 women presented for breech birth planning. ECV was performed on 353 patients. The overall success rate was 22.4%. A later week of gestation (odds ratio [OR] 1.69), an abundant amniotic fluid index (AFI score) (OR 5.74), fundal (OR 3.78) and anterior (OR 0.39) placental location, and an oblique lie (OR 9.08) were significantly associated with successful ECV in our population. No major complications were observed. The overall vaginal delivery rates could be increased to approximately 14% with ECV.

### Conclusion

The demand for alternative birth modes other than CS for breech birth is high in the area of Frankfurt, Germany. Our study offers evidence of the safety of ECV at 38 weeks. Centers with expertise in vaginal breech delivery and ECV can reduce CS-rates. To further establish

**Data Availability Statement:** The minimal data set is within the paper. Additional data may be available upon request.

**Funding:** The authors received no specific funding for this work.

**Competing interests:** Prof. Frank Louwen is first vice president of the German Society for Gynaecology and Obstetrics (DGGG), council member of the European Board and College of Obstetrics and Gynaecology (EBCOG) and Executive Board Member und Committee chairman of the International Federation of Gynecology and Obstetrics (FIGO). Prof. Frank Louwen und Dr. Cover Letter Lukas Jennewein worked on the first German S3-Guideline for cesarean section published in 2020. Ann-Sophie Zielbauer declares no relevant conflicts of interest. This does not alter our adherence to PLOS ONE policies on sharing data and materials.

vaginal breech delivery and ECV as alternate options, the required knowledge and skill should be implemented in the revised curricula.

## Introduction

The cesarean section (CS) rate is increasing worldwide, and surpassing 50% of all births in some countries [1]. Since CS is associated with severe complications, increasing CS rates contribute to a rise in maternal mortality worldwide, with a mortality rate of 8/1,000 for procedures in low- and middle-income countries and 16/100,000 per birth in more developed countries [1–3]. In Germany, 30.5% of all babies are born through CS [4]. The fetal breech presentation is among the most critically discussed indications for a CS. In Germany, 65.7% of pregnant women with breech presentation receive planned CS at term [5]. The CS rate is over 90% in some countries [6].

Vaginal breech births are a possible alternative. Unfortunately, the expertise for vaginal breech delivery has rapidly declined over the last two decades, according to the study by Hannah et al. [7]. In nationwide guidelines and committee opinions, it is (1) proposed to be a safe option to deliver vaginally and (2) recommended to offer external cephalic version (ECV) to patients with breech presentation [8,9]. Previous studies have shown that vaginal breech delivery at term is not accompanied by increased maternal or infant morbidity, even with a high fetal weight [10].

The aim of ECV is to rotate the fetus, resulting in a vertex position, by manipulation through the maternal abdomen. Even though ECV is recommended based on current gynaecologic guidelines, it is not always offered in Germany or is refused by pregnant women [11,12]. Little information is available regarding the demographics and implementation rates in Germany. In a recently published multicenter observational study in Germany, hospitals were questioned about breech birth and the ECV approach. Unfortunately, the response rate was low (37.2%) [12].

The ideal week of gestation for ECV has been investigated in multiple studies. The current data situation was heterogeneous. Most studies comparing an early attempt at 36–37 weeks of gestation with a late attempt at 37–38 weeks of gestation, showed a higher success rate for ECV at 36–37 weeks, accompanied by a higher risk for preterm birth [13–17]. In contrast, a recently published large cohort study demonstrated equal success and preterm birth rates regardless of the week of gestation [18]. The American College of Obstetricians and Gynecologists and the Royal College of Obstetricians and Gynecologists guidelines recommend that ECV should be performed beginning at 37+0 weeks to decrease the rate of reversion and increase the rate of spontaneous version [8,19]. To the best of our knowledge, many obstetrics departments in Germany offer ECV before 37 weeks of gestation [12,20].

The Frankfurt University Hospital offers vaginal breech delivery to all women presenting for breech consultation and has the highest number of ECVs and vaginal breech delivery in the federal state. With the offer of breech delivery in an upright position, we were able to further decrease CS rates by 32% [21].

We aimed to evaluate whether the routine offer of ECV at 38 weeks of gestation can further reduce CS rates at a center specializing in vaginal breech birth. Therefore, this study primarily aimed to analyze the ECV success rate and delivery outcomes at a tertiary obstetrics center in Germany. Our secondary aim was to analyze the prognostic factors of ECV, our patient characteristics, and birth modalities along with complications and perinatal outcomes.

## Materials and methods

We conducted a prospective analysis of all women presenting for counseling with a singleton fetus in breech presentation after 34 weeks of gestation at a tertiary healthcare center for obstetrics between January 2013 and December 2017. Patients with multiple pregnancies were excluded from the study. A study period of 5 years was chosen in order to arrive at a representative sample size. The ethics committee of the Goethe University Hospital Frankfurt, approved the study protocol (ref: 176/18). Patient consent was waived because we analyzed routinely collected medical data. All data were retrospectively gathered after patient discharge.

At our department, women with a fetus in breech presentation were recommended to present themselves after 34 weeks of gestation for birth counseling. ECV is offered to all patients with a singleton breech pregnancy in the absence of contraindications. Contraindications include intrauterine growth restriction, fetal malformations, uterine myomatosus, and placental or uterine abnormalities. Ultrasound examination was performed, and fetal weight, type of breech, placental location, and amount of amniotic fluid were documented within the standard counseling procedure.

Two doctors performed ECV together monitoring of fetal heart rate using ultrasonography. Fenoterol was used as an intravenous uterine relaxant starting 30 min prior to the procedure and was continuously applied until completion of the procedure. The baby was moved upwards with one hand and pushed to perform a forward or backward roll, preferably in the direction with less resistance. In cases of unsuccessful ECV (NECV), both directions were attempted. The direction (backwards or forwards) of successful ECV (SECV) has not been documented.

Patients presenting for breech birth planning were registered and abstracted for ECV eligibility and birth mode. Women giving birth at another center were lost to follow-up. All patients who underwent ECV at our center were included in the analysis. For these patients, maternal patient history (age, height, weight, underlying diseases, number of pregnancies, and childbirth) and fetal biometrics (type of breech, placental location, and amount of amniotic fluid) were extracted and compared. Fetal weight was estimated sonographically (by Hadlock), and the amniotic fluid index was measured according to Phelan et al. [22,23]. An AFI $\leq 7$ was defined as a scarce amount of amniotic fluid, and an AFI $\geq 20$, an abundant amount. All data regarding the ECV procedure were documented. For further analysis, births after SECV and NECV were examined separately. In women giving birth at our clinic after ECV, the birth mode and outcome parameters of the mother and neonates were additionally analyzed. Routine patient history documentation was used to select and extract the data.

Statistical analysis was performed using Excel, BiAS v11.08, and IBM SPSS Statistics 22. For descriptive analysis, means, medians, and percentages were calculated using variance and standard deviation. Confidence intervals were calculated at 95%, and p-values were calculated bilaterally, with statistical significance set at $p < 0.05$. As missing data occurred without pattern, a complete case analysis was performed for all variables.

We used the binary outcome (success or failure) of the ECV as a grouping variable and tested all variables for significance using the chi-square test for nominal variables, the Mann–Whitney U test for ordinal variables, and an independent samples t-test for all continuous variables. Then, a Bonferroni correction was applied, and a logistic regression with backward elimination was carried out for all significant variables to confirm the findings.

## Results

Within the observation period, a total of 1,598 patients with breech presentation visited the consultation center for birth planning. We observed an average yearly increase in

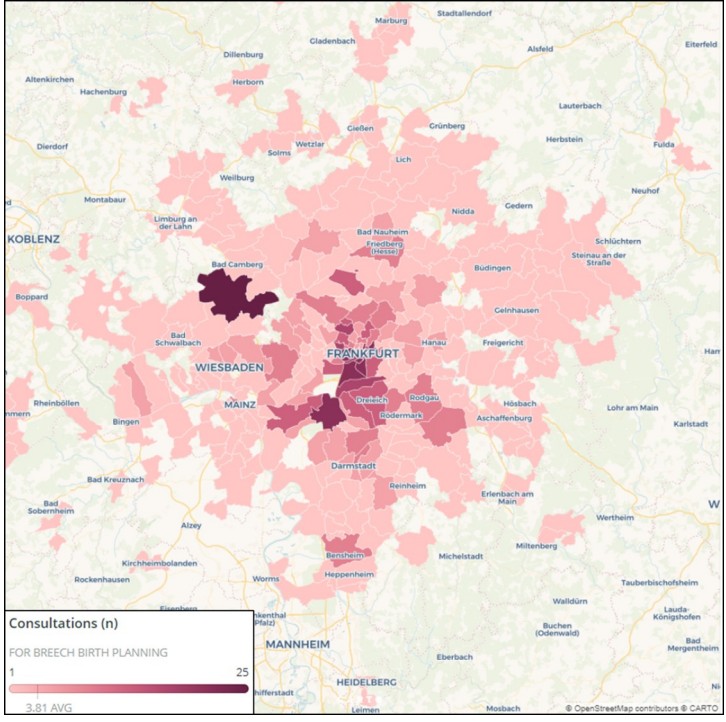

**Fig 1. Hospital catchment area.** Hospital catchment area for outpatient consultations for birth planning with breech presentation. Patients from areas outside of the depicted map were excluded. The number of consultations over 5 years is depicted for each zip code. Color was chosen at equal intervals. The map was created using CARTO and OpenStreetMaps. Reprinted from https://carto.com under a CC BY license, with permission from CARTO Legal, original copyright 2021.

consultations of 9.0% over the 5-year period, resulting in a total increase from 265 patients in 2013 to 366 in 2017.

The catchment area for outpatient consultation is shown by postal code in Fig 1. The highest incidence of consultation was observed in close proximity to the department. The catchment area exceeds Frankfurt by up to 200 km despite neighboring obstetric departments in Darmstadt, Wiesbaden, and Mainz. Forty percent of our patients presenting for breech birth planning have a residence exceeding a 30-km radius.

Of the consulting patients, 1,398 (87.5%) women made follow-up appointments and completed the preceding diagnostics for ECV and breech delivery. A total of 61.5% would have been suitable for ECV. Twelve children were born premature prior to the ECV appointment. A total of 381 women presented with ECV at our department, and ECV was attempted in 353 women. Moreover, 28 fetuses had spontaneously turned into cephalic presentation prior to the procedure.

A flowchart of outpatient consultations for birth planning with breech presentation is shown in Fig 2.

Of the 353 women undergoing ECV, the median gestational age at ECV was 37+5 weeks, and the majority were nulliparous (70.8%). Most fetuses were in frank breech presentation, mostly with a posterior placental location. The median estimated birth weight at the ECV was 3,260 g. The overall success rate for ECV was 22.4%. Maternal and fetal characteristics for SECV and NECV are compared in Table 1.

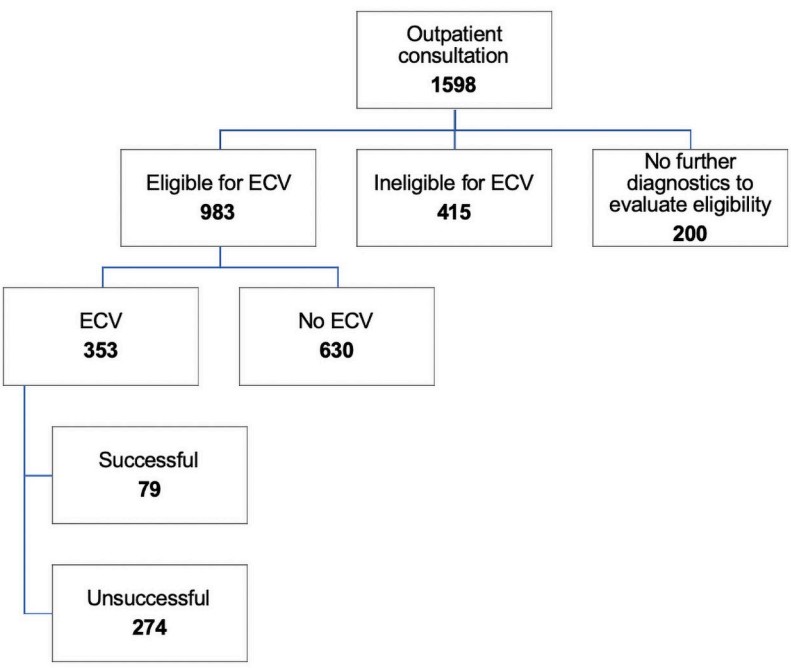

**Fig 2. Flowchart of outpatient consultations for birth planning with breech presentation.**

The outcomes differed significantly between SECV and NECV in terms of parity, gestational age, estimated fetal weight, breech type, amniotic fluid index, and placental location.

In nulliparous women, the success rates were 18.4% and 32.0% in women with multiple pregnancies. The amniotic fluid index was significantly higher among patients with SECV. The amniotic fluid index ranged from 3.3 cm to 23 cm.

For the SECV group, the fetal birth weight was higher (2,959 g) compared to that in the NECV group (2,860 g, p = 0.028), as well as the biparietal diameter (BPD) (94.3 mm vs. 92.9 mm, p = 0.35). Gestational age was higher for SECV.

In cases of placental location, SECV was associated with a fundal placental position, whereas the prevalence of an anterior location was significantly higher in NECV. For the type of breech, SECV was significantly associated with an oblique lie, whereas NECV was significantly associated with frank breech presentation.

After logistic regression with stepwise backward elimination, the AFI score, placental location, and type of breech remained significant in the final model.

The oblique lie had a significant and tenfold chance for SECV than any other breech presentation (p = 0.003); an abundant AFI score increased the chances of SECV by almost six times (p = 0.037). The only variable with a significant negative impact on SECV was the anterior placental location (odds ratio [OR] 0.39; p = 0.003). The estimated birth weight, frank breech presentation and multiparity were not significant. The adjusted ORs with confidence intervals are shown in Table 2.

Within the first 24 h after ECV, complications were noted in 60 (17.0%) patients. Most complications were nonpermanent. Complications were divided into minor and major complications, as shown in Table 3. Minor complications, such as cardiotocography (CTG) alterations, had no clinical significance. Major complications were defined as complications leading to a measurable impact on the mother or fetus. We reported the most severe

**Table 1. Maternal and fetal characteristics.**

| Maternal Characteristics | Total N = 353 | SECV N = 79 | NECV N = 274 | p-Value |
|---|---|---|---|---|
| | Mean ± SD | Mean ± SD | Mean ± SD | |
| Maternal age, y | 32.6 ± 4.3 | 33.1 ± 4.2 | 32.5 ± 4.3 | 0.17 |
| Maternal height, cm | 169.2 ± 6.0 | 168.9 ± 5.9 | 169.3 ± 6.0 | 0.62 |
| Maternal weight, kg | 65.8 ± 12.9 | 63.7 ± 9.2 | 66.5 ± 13.7 | 0.23 |
| Gestational age at ECV, wk | 37.2 ± 0.6 | 37.4 ± 0.71 | 37.2 ± 0.5 | 0.001* |
| Gravida | 1.6 ± 1.0 | 1.9 ± 1.0 | 1.6 ± 1.0 | 0.013* |
| Para | 0.4 ± 0.71 | 0.6 ± 0.8 | 0.3 ± 0.7 | 0.004* |
| Conjugata obstetrica, cm | 12.8 ± 0.8 | 12.9 ± 0.8 | 12.6 ± 0.8 | 0.156 |
| **Fetal Characteristics** | | | | |
| **Placental location** | | | | |
| Anterior | 146 (41.4%) | 20 (25.3%) | 126 (46.0%) | 0.001* |
| Posterior | 163 (46.2%) | 43 (54.4%) | 120 (43.8%) | 0.091 |
| Fundal | 16 (4.5%) | 8 (10.1%) | 8 (2.9%) | 0.007* |
| Left | 15 (4.3%) | 4 (5.1%) | 11 (4.0%) | 0.62 |
| Right | 9 (2.6%) | 3 (3.8%) | 6 (2.2%) | 0.42 |
| Missing data | 4 (1.1%) | 1 (1.3%) | 3 (1.1%) | |
| **Type of breech** | | | | |
| Complete | 56 (15.9%) | 12 (15.2%) | 44 (16.1%) | 0.85 |
| Incomplete | 38 (10.8%) | 9 (11.4%) | 29 (10.6%) | 0.84 |
| Double footling | 5 (1.4%) | 1 (1.3%) | 4 (1.5%) | 0.90 |
| Frank | 210 (59.5%) | 38 (48.1%) | 172 (62.8%) | 0.019* |
| Oblique lie | 12 (3.4%) | 8 (10.1%) | 4 (1.5%) | 0.000* |
| Breech, not further specified | 32 (9.1%) | 11 (13.9%) | 21 (7.7%) | |
| **Amount of amniotic fluid** | | | | |
| Scarce | 32 (9.1%) | 3 (3.8%) | 29 (10.6%) | 0.07 |
| Normal | 311 (88.1%) | 71 (89.9%) | 240 (87.6%) | 0.49 |
| Abundant | 7 (2.0%) | 4 (5.1%) | 3 (1.1%) | 0.025* |
| Missing data | 3 (0.9%) | 1 (1.3%) | 2 (0.7%) | |
| | Mean ± SD | Mean ± SD | Mean ± SD | |
| **Estimated fetal weight, g** | 2881.8 ± 333.9 | 2958.9 ± 349.8 | 2860.1 ± 349.8 | 0.028* |

Maternal characteristics of all women undergoing ECV and fetal characteristics prior to ECV. All ratios in the first column are listed as a percentage of the total fetal number, in the second column as a percentage of all successful ECVs and in the third column as percentage of all not successful ECVs. All metric parameters are presented as means and standard deviations. T-test of the mean difference between SECV and NECV.

**Table 2. Logistic regression.**

| Variable | Odds Ratio | 95% CI | Wald's p |
|---|---|---|---|
| Week of gestation | 1.69 | 1.08–2.65 | 0.021 |
| Anterior placental location | 0.39 | 0.21–0.73 | 0.003 |
| Fundal placental location | 3.78 | 1.26–11.40 | 0.018 |
| Abundant AFI score | 5.74 | 1.11–29.55 | 0.037 |
| Oblique lie | 9.08 | 2.10–39.25 | 0.003 |

The variables remaining in the final model after backward selection are presented. All variables significantly associated with SECV were included in the logistic regression analysis.

**Table 3. Complications.**

| Complications | N | % of all complications | % of all ECVs | Complication leading to CS |
|---|---|---|---|---|
| **Minor complications** | **43** | **71.7%** | **12.2%** | **0** |
| Bradycardia of the fetus | 25 | 41.7% | 7.1% | 0 |
| Other CTG alterations | 8 | 13.3% | 2.3% | 0 |
| Vaginal blood loss | 2 | 3.3% | 0.6% | 0 |
| Loss of amniotic fluid[a] | 1 | 1.7% | 0.3% | 0 |
| Non-persisting contractions | 4 | 6.7% | 1.1% | 0 |
| Other complications | 3 | 5.0% | 0.9% | 0 |
| **Major complications** | **15** | **25.0%** | **4.3%** | **8** |
| Rupture of membranes | 7 | 11.7% | 2.0% | 3 |
| Contractions with onset of birth | 6 | 10.0% | 1.7% | 4 |
| Other CTG alterations leading to premature birth | 1 | 1.7% | 0.3% | 0 |
| Vaginal blood loss | 1 | 1.7% | 0.3% | 1 |
| **Complications leading to emergency CS** | **2** | **3.3%** | **0.6%** | **2** |
| Placental abruption | 1 | 1.7% | 0.3% | 1 |
| Persisting bradycardia of the fetus | 1 | 1.7% | 0.3% | 1 |
| **Total number of patients with complications** | **60** | | **17.0%** | **10** |

Complications were divided into minor and major complications. Minor complications are nonpersistent and do not lead to the onset of labor, whereas major complications lead to the onset of birth within 24 h. Major complications leading to emergency CS are presented separately.

[a]One case of temporal loss of vaginal fluid by drop.

complication in six patients with multiple complications (Table 3). The most frequent fetal complications were CTG alterations, especially bradycardia. Most CTG alterations (85.7%) under ECV were nonpermanent and terminated when the procedure was stopped. In five cases, tocolytics had to be administered to end fetal bradycardia. Clinical significant pain or maternal circulatory problems leading to an early termination of ECV are rare, accounting for 1.4% of all ECVs. Five procedures were stopped upon patient demand or difficulties due to obesity.

Labor began within 24 h after ECV in 4.8% of cases. In 41.2% of cases, the child could be delivered vaginally. Emergency CS was performed twice, once due to placental abruption immediately after ECV, and once due to therapy-resistant bradycardia of the fetus. Three children were admitted to the newborn ICU because of respiratory adaption disorder or newborn infection. At discharge from the hospital, all infants were healthy and clinically stable.

A total of 252 (71%) patients attending ECV also delivered in our department. Patients were more likely to deliver elsewhere after SECV (59.5%), while the majority of patients after NECV (80.3%) were delivered at our center. Successful vaginal delivery after ECV was achieved in 62.3% of all women delivered to our clinic. After SECV, only one fetus had turned into an oblique lie, and all the others were born out of cephalic presentation. A total of 78.13% of all women with SECV delivered vaginally. After the unsuccessful version, the chances for a spontaneous version were low (2%). Two fetuses had turned spontaneously into cephalic presentation, and three were in oblique lie. Vaginal delivery was attempted in 82.6% of all women with NECV delivery at our center. Further, 59% of these successfully delivered vaginally out of breech presentation. In 23.6% of cases, a CS under labor had to be performed due to obstructed labor or pathological CTGs. A total of 13.2% in the NECV group decided to undergo elective CS.

**Table 4. Characteristics of birth.**

| Characteristics of Birth | All Deliveries after ECV N = 252 | Deliveries after SECV N = 32 | Deliveries after NECV N = 220 |
|---|---|---|---|
| **Presentation at birth** | | | |
| Breech | 215 (85.3%) | 0 (0%) | 215 (97.7%) |
| Cephalic presentation | 33 (13.1%) | 31 (96.9%) | 2 (0.9%) |
| Oblique lie | 4 (1.6%) | 1 (3.1%) | 3 (1.4%) |
| **Delivery** | | | |
| Vaginal birth in total | 157 (62.3%) | 25 (78.1%) | 132 (60.0%) |
| Vaginal breech birth | 130 (51.6%) | 0 (0%) | 130 (59.1%) |
| Elective CS | 30 (11.9%) | 1 (3.1%) | 29 (13.2%) |
| CS after onset of labor | 58 (23.0%) | 6 (18.8%) | 52 (23.6%) |
| Emergency CS | 7 (2.8%) | 0 (0%) | 7 (3.2%) |
| **Difference in discharge dates between mother and child** | 0.2 ± 1.4 | 0.3 ± 1.4 | 0.2 ± 1.4 |
| **Child characteristics** | | | |
| Male sex | 117 (46.4%) | 17 (53.1%) | 100 (45.5%) |
| Female sex | 135 (53.6%) | 15 (46.9%) | 120 (54.6%) |
| Birth weight | 3,333.9 ± 423.2 | 3,511.1 ± 376.8 | 3,308.1 ± 423.8 |
| Fetal weight, g | 2,881.8 ± 333.9 | 2,958.9 ± 324.3 | 2,860.1 ± 326.7 |
| Height (cm) | 51.9 ± 2.8 | 52.2 ± 1.8 | 51.89 ± 2.8 |
| Head circumference (cm) | 35.5 ± 1.5 | 35.8 ± 1.4 | 35.5 ± 1.4 |
| **Child outcome** | | | |
| pH < 7 | 5 (2.0%) | 0 (0%) | 5 (2.3%) |
| 5-min Apgar score < 5 | 1 (0.4%) | 0 (0%) | 1 (0.5%) |
| BE < -12 | 13 (5.2%) | 1 (3.1%) | 12 (5.5%) |
| **Intensive care unit child** | | | |
| Number of transfers | 19 (7.5%) | 1 (3.1%) | 18 (8.2%) |
| Respiratory adaptation disorder | 6 (2.4%) | 0 (0%) | 6 (2.7%) |
| Newborn infection | 11 (4.4%) | 1 (3.1%) | 10 (4.6%) |
| Other | 2 (0.8%) | 0 (0%) | 2 (0.9%) |

The characteristics of birth of all women who delivered in our clinic after ECV. All ratios in the first column are listed as a percentage of the total number, in the second column as a percentage of all successful ECVs and in the third column as a percentage of all NECVs. All metric parameters are presented as means and standard deviations.

Fetal outcomes did not differ significantly between the two groups (APGAR < 5; 0% vs. 0.5%, p = 0.072; base excess < -12; 3.1% vs. 5.5%, p = 0.57 or pH of the umbilical artery < 7; 0% vs. 2.27%, p = 0.39). There was no significant difference in the rate of admission to a neonatal intensive care unit (NICU) or the difference in discharge dates between mothers and children. There were 19 children transferred to a NICU. The most common reason was newborn infection. Six children had congenital abnormalities, such as congenital heart disease. In 4.8% of the discharge dates, the mother and child differed by a maximum of 13 days. The delivery information and child outcomes are shown in Table 4.

A total of 845 women who decided against ECV still delivered in our center. Of these, 7.0% had spontaneously turned into cephalic presentation before birth. A total of 573 women underwent a vaginal breech delivery. The birth mode without prior ECV is shown in Fig 3. We achieved a total vaginal delivery rate of 50.9% of all patients presenting for breech birth planning and delivered in our center disregarded ECV.

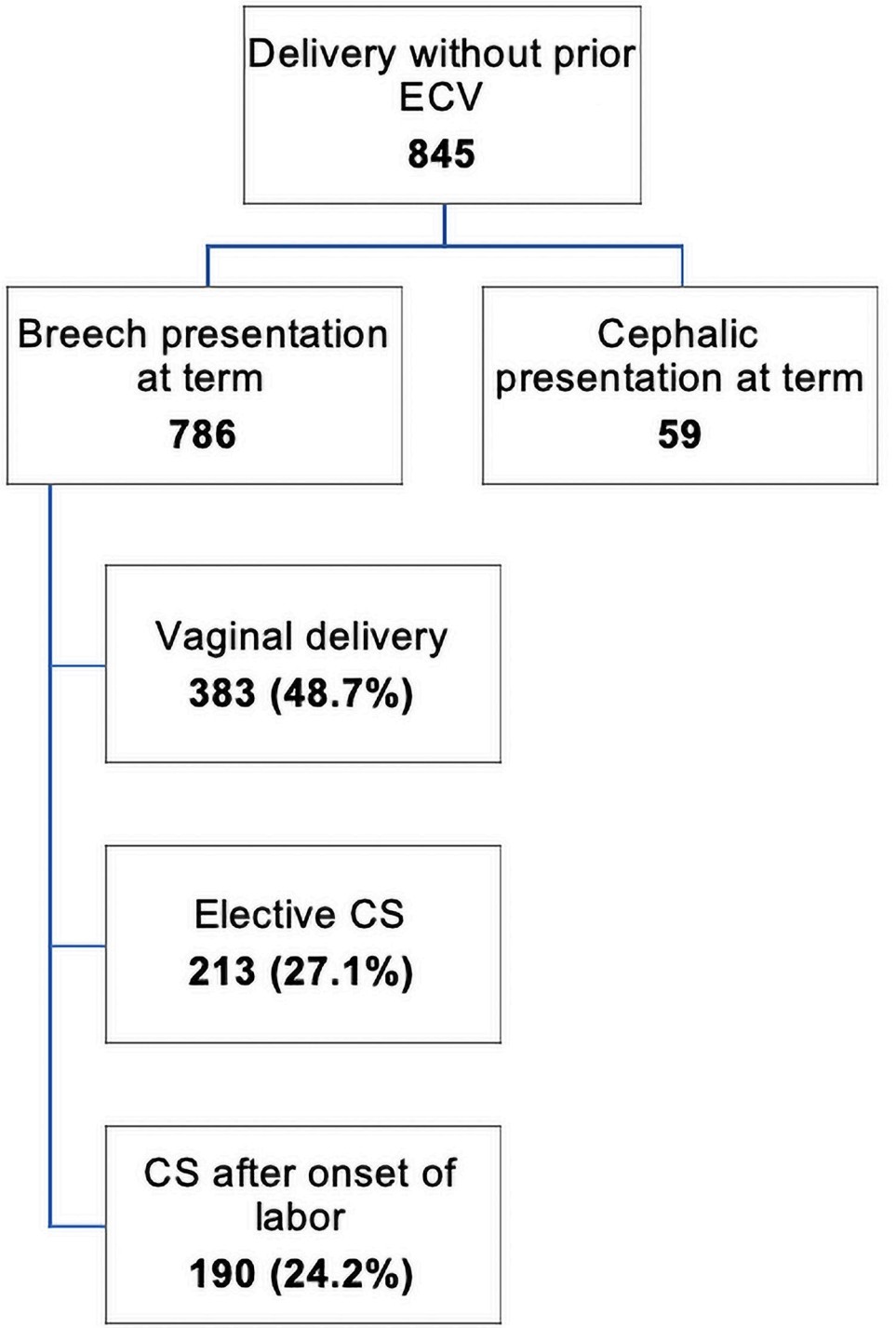

**Fig 3. Flowchart showing the delivery mode of all women without prior ECV and delivered in our center.**

## Discussion

### Main findings

Over the 5-year period, we observed a low uptake rate for ECV at 22.1% and a high rate for breech vaginal delivery with and without prior ECV of over 50%. The overall success rate for ECV at 38 weeks (37+0 to 38+0) was 22.4%. A significant correlation between SECV and the gestational age, an abundant AFI score, fundal placental location, and an oblique lie can be seen. Conversely, an anterior placental location was significantly associated with NECV. No major complications related to ECV resulting in maternal or fetal morbidity or mortality were observed over the 5-year period.

### Strength and limitations

Our study is the first to analyze the impact of an ECV in a highly experienced center for vaginal breech birth. In many centers, vaginal breech delivery is not as equally offered as an ECV [12]. ECV then constitutes the only possibility for vaginal delivery and decreases the likelihood of CS. At our center, we offer vaginal breech delivery as an equal alternative to breech management. Our consultation for breech birth planning follows a standardized procedure, in which all possible procedures such as ECV, vaginal breech birth, and CS were discussed, to reach a participatory decision with the patient. Observing for five years, we included a large number of patients who consulted for breech birth planning.

The major limitation of this study is the low number of SECVs, which resulted from the relatively low uptake and success rates. These rates resemble our highly specialized center for vaginal breech delivery. In contrast, we reported a high number of breech deliveries. We were able to show that a SECV is not the only option for vaginal birth. Our data differ from those of previous studies especially because of the high rate of breech delivery.

Therefore, our findings might not be transferable to other hospitals or countries with different levels of knowledge and experience in ECV and vaginal breech delivery.

By performing ECV at 38 weeks risks such as preterm birth were reduced, and security of the procedure can be provided, which is crucial for our consultation offering the safest options for mothers and children.

Another limitation is the lack of information on women giving birth in other hospitals. The loss to follow-up was especially high in women with SECV. Conversely, one aim of ECV is the opportunity for women to have a cephalic birth close to their home.

### Interpretation

A few international studies have reported ECV uptake rates ranging from approximately 20% to 70% in the Netherlands [11,20,24,25]. Compared with other centers, we encountered a low uptake rate for ECV and a high uptake rate for vaginal breech delivery. Most women eligible for ECV waived the offer and decided for vaginal breech delivery, resulting in a relatively low uptake rate for ECV. Other hospitals in the catchment area offer ECV but do not routinely offer vaginal delivery out of the breech presentation. Pregnant women who wish to deliver vaginally when their baby is in breech presentation tend to consult in our center. Women preferring a CS from the beginning as the mode of delivery out of the breech presentation are probably more likely to choose other obstetrical departments for consultation, ECV, or cesarean delivery leading to lower ECV uptake rates at our center.

Furthermore, women in Germany are still reluctant to use ECV as a procedure. The fear of pain or discomfort is the main reason for choosing ECV [11,26]. If the resident gynecologist has already advised against ECV and vaginal breech birth, and advocated for CS, pregnant

women may not reconsider this option. These recommendations by resident doctors against ECV and vaginal breech birth may still be based on the breech term trial and based on false risk assumptions [7].

However, the routine use of ECV resulted in an increase in vaginal delivery. Compared to the international literature, we observed a higher rate of vaginal delivery after NECV (15.5% described by Trobo et al. vs. 66.8% in our population) [27]. Our total rate of vaginal delivery exceeds the German average of 6.75% vaginal delivery from breech presentation by far [5]. ECV and vaginal breech delivery should be offered to all applicable patients, as already manifested in many obstetrician guidelines [28].

Our overall success rate is within the lower end of the internationally reported range [29,30]. As recommended in international and national guidelines, ECV has been performed at our center at 38 weeks of gestation in the past decade. Previous multicenter studies have compared ECVs prior to term and at term. These studies indicated that ECV success rates might be higher when ECV is performed before 36 weeks of gestation, but may also increase the risk of preterm birth [14,15]. To decrease the risk of preterm birth, we accept a lower success rate for ECV by offering ECV at 38 weeks of gestation. Another reason for the lower success rate might be that ECV attempts in women considering vaginal breech delivery could be less intense carried out compared to that in women who wish to deliver vaginally out of cephalic presentation.

Obstetricians in our center also attempt ECV despite little chance for success due to the patient's wishes, for example in cases of a scarce AFI-score, which has already been described to be correlated with a NECV [17,20]. Nulliparity has been associated with a lower ECV success rate [31]. The proportion of nulliparous women was lower in our population than in most studies, with approximately 55% nulliparity [27,32,33]. This could partially explain the lower success rates at our center. If ECV was only performed in patients with preferential prognostic conditions, the success rate might have been higher. However, stricter exclusion criteria would further decrease the total amount of ECVs.

Drug interventions that improve the ECV success rate include neuraxial analgesia and tocolytics. The impact of beta-stimulant tocolytics, calcium channel blockers and oxytocin antagonists on the success rate of ECV has already been studied. All drugs have been shown to be safe with only few side effects, but beta-stimulants significantly improved the success rate compared to other tocolytics [27,34–36]. In our clinic, fenoterol is a standard tocolytic drug for ECV. The use of a different tocolytic drugs might not improve the ECV success rates in our population.

A systematic review by Magro-Malosso et al. showed a significant increase in the success rate of ECV by administering neuraxial analgesia in addition to tocolytics [37]. Whether women with low success rates, such as nulliparous women, would benefit more from neuraxial analgesia than multiparous women, where success rates are already higher, should be the subject of further research.

Factors such as multiparity, posterior placental location, and a high AFI score have been previously described to increase the success rate of ECV in most studies. In contrast, nulliparity, anterior placental location, and oligohydramnion were described to reduce the chances of success [20,27,31–33,38,39]. Within our population we were able to confirm the negative impact of an anterior placental location, as well as the positive impact of a high AFI score. For multiparity, we were able to show a positive impact on the success rate, but we were not able to statistically confirm this parameter. For posterior placental location or scarce amniotic fluid index, no statistical impact on the success rate could be confirmed within our population.

Other parameters such as breech presentation, maternal body mass index (BMI), and gestational age are described to influence ECV as well, but the data situation is heterogeneous

[20,33,38]. Within our population, oblique lie was significantly associated with SECV, which was previously described by Salzer et al. [40]. For maternal height and weight, we found no significant effect on the success rate. The influence of these parameters might be of minor importance or may be correlated with other maternal or fetal factors.

In Germany, a commonly used score for the evaluation of success is the score by Kainer et al. [12,41]. It was established for ECV at 36 weeks of gestation with prognostic factors such as an AFI score ≥7, posterior placental location, multiparity, and a lower estimated fetal weight. We were able to confirm these parameters for 38 weeks of gestation, except for fetal weight. In our population, we were surprisingly able to show an inverse effect on gestational age with a supporting tendency for higher birth weight. In a recently published paper success rates were significantly higher in children born with a birth weight of 3.5 kg or above [18]. Hakem et al. assumed that a larger fetus is less engaged in the pelvis and is therefore easier to rotate. Another possible reason might be that the larger fetus is more palpable, leading to a greater power transmission, which results in a higher mobility of the fetus.

When ECV is performed at 38 weeks of gestation, it might be recommended to attempt ECV at the end of 38 weeks. With this modification, the score is applicable for ECV at 38 weeks.

ECV has been described as a safe procedure [42]. We were able to confirm that ECV at 38 weeks of gestation was not associated with higher risks for the mother or child. Most of the observed complications were non-severe. All complications defined as minor, such as short bradycardias or other CTG alterations were not clinically significant, but led to a higher number of complications in total than reported in previous studies [32,43,44]. Grootscholten et al. recommended only considering cardiotocographic abnormalities leading to CS as a complication of the ECV. [43] Beuckens et al. only reported on more severe complications when an obstetrician was consulted, for example, bradycardia lasting over 10 min [32]. Our complication rate might seem higher at first sight, but comparing serous events such as preterm birth or placental abruption, our complication rate is comparable to those described by Grootscholten et al. or Beuckens et al. [32,43]. We observed no of the other major complications as described by Rodgers et al., such as bone fracture, cord prolapse or fetal death [44].

Considering our large catchment area for breech consultation, there is a high demand for alternatives to planned CS when breech presentation is apparent in Germany. In a recently published multicenter study, Kohls et al. described the majority of hospitals offering ECV in Germany as university or maximum care hospitals [12]. One reason might be that our expired German guideline for breech presentation only comments on ECV in an annex stating that it may be carried out [45]. This might change in the near future because the recently published German guideline on CS provides a stronger recommendation on ECV [46].

The rate of elective CS for women presenting with a breech presentation at term at our center is lower than the nationwide average (23.8% *vs.* 65.7%) [5]. This underlines the necessity of offering an ECV. However, a significant number of CSs were performed on patient demand. Two out of three women with planned CS had no contraindications for vaginal breech delivery. Independent information and early evidence-based counseling may further decrease CS numbers.

## Conclusion

By offering both ECV and vaginal breech delivery, we were able to give women with breech pregnancy one more possibility for breech delivery. In our cohort, the vaginal birth approach in breech presentation was often preferred in pregnant women seeking consultation in both women with and without prior ECV. Despite our low success rates for ECV, a SECV was

effective at increasing the rate of vaginal birth. Obstetricians should always discuss ECV as a first-step approach to breech presentation. In particular, by offering ECV at 38 weeks, a thorough risk-benefit analysis has yielded positive results. It is desirable that ECV and vaginal breech deliveries are further implemented in the clinical routine of obstetricians in Germany and spread methods and clinical competence through practical guided training so that more centers are able to offer further consultation and clinical management. Especially in the primary care center, thorough patient education on ECV and vaginal breech delivery can further decrease barriers to the procedure and help patients during the decision-making process on their optimal form of delivery.

We were able to confirm the previously described prognostic factors and the safety of ECV. For future consultations, it might be useful to further implement these in the consultation process and recommend women with unfavourable chances of ECV a vaginal breech birth.

## Acknowledgments

We thank all involved team members at our university hospital (midwives, nurses, doctors, and other hospital employees). We are grateful to all participating patients.

## Author Contributions

**Conceptualization:** Frank Louwen, Lukas Jennewein.

**Data curation:** Ann-Sophie Zielbauer.

**Formal analysis:** Ann-Sophie Zielbauer, Lukas Jennewein.

**Methodology:** Ann-Sophie Zielbauer, Lukas Jennewein.

**Project administration:** Frank Louwen.

**Supervision:** Frank Louwen, Lukas Jennewein.

**Validation:** Lukas Jennewein.

**Visualization:** Ann-Sophie Zielbauer.

**Writing – original draft:** Ann-Sophie Zielbauer.

**Writing – review & editing:** Ann-Sophie Zielbauer, Frank Louwen, Lukas Jennewein.

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
