## [Decision Letter · Decision Letter 0]

29 Sep 2020

PONE-D-20-24091

External cephalic version in 38 weeks’ gestation – demographics, prognostic factors and success rates: a prospective analysis from a German single center

PLOS ONE

Dear Authors,

Thank you for submitting your manuscript to PLOS ONE. After careful consideration, we feel that it has merit but does not fully meet PLOS ONE’s publication criteria as it currently stands. Therefore, we invite you to submit a revised version of the manuscript that addresses the points raised during the review process.

We look forward to receiving your revised manuscript.

Kind regards,

Salvatore Andrea Mastrolia, M.D.

Academic Editor

PLOS ONE

Journal Requirements:

'I have read the journal's policy and the authors of this manuscript have the following competing interests:

F.L. is first vice president of the German Society for Gynaecology and Obstetrics (DGGG), council member of the European Board and College of Obstetrics and Gynaecology (EBCOG) and Executive Board Member und Committee chairman of the International Federation of Gynecology and Obstetrics (FIGO).

F.L. and L.J. worked on the first German S3-Guideline for cesarean section published in 2020.

A.Z. declares no relevant conflicts of interest.'

a. Please confirm that this does not alter your adherence to all PLOS ONE policies on sharing data and materials, by including the following statement: "This does not alter our adherence to  PLOS ONE policies on sharing data and materials.” (as detailed online in our guide for authors http://journals.plos.org/plosone/s/competing-interests).  If there are restrictions on sharing of data and/or materials, please state these.

Please note that we cannot proceed with consideration of your article until this information has been declared.

Reviewers' comments:

Reviewer's Responses to Questions

**Comments to the Author**

1. Is the manuscript technically sound, and do the data support the conclusions?

Reviewer #1: Partly

Reviewer #2: Partly

2. Has the statistical analysis been performed appropriately and rigorously? 

Reviewer #1: No

Reviewer #2: Yes

3. Have the authors made all data underlying the findings in their manuscript fully available?

Reviewer #1: Yes

Reviewer #2: Yes

4. Is the manuscript presented in an intelligible fashion and written in standard English?

Reviewer #1: No

Reviewer #2: Yes

5. Review Comments to the Author

Reviewer #1: title - too long

keywords - too many

introduction - ECV is mostly performed before 37 GA?

LINE 67 - provide reference for most ecv performed before 37

line 68-70 - is redundant

ECV not offered line79 - provide reference

line 79-81 - more studies to be published??? numerous studies regarding predictors of success. the authors should replace outofdate reference (years 2000-2010) regarding success rate (ezra, ben meir) and replace with more updated reference (31312960, 30941816)

a prospective analyses? is this trully prospective or retrospective? approval (Ref:176/18) means 2018?

extremely low success rate of ECV -22% should be discussed and explained

reults chapter is written badly - please rewrite.

tables - ages with 2 decimals? what age does 33.13 represents?? parity and gravidity with decimals?

table 2 looks like it was copied from the SPSS sowtware - please provide a better version of presentation of table

what does constant mean?

vaginal delivery rate of 62% is very low - should be discussed

line 275-276 claiming novely - what about 31312960?

overall i dont see novelty, maybe unfamiliarity with current literature, therefore a more thorough literature review should be performed in the introduction and discussion and old references (more than 10 years should be discuraged)

limited new information, low success rate - thus limiting generalizability

Reviewer #2: First of all, this is a very interesting manuscript about management of breech presentation at term. This Care Unit seems to be a very expertise Breech Unit, and provide care about counseling about all possible maneuvers such as ECV, cesarean and vaginal birth. Congratulations for that.

However, it is commented on several occasions that this is the first article published about the safety of ECV in 38 weeks of gestation (lines 39, 49, 85, 275). The study fails to address how the findings relate to previous research in this area. The authors should return to publications such as that of Rodgers et al., 2017 (reference number 32 of the manuscript), where the mean gestational age of the ECV was 37 + 5, or in others such as Beuckens et al., 2015 (an observational study of the success and complications of 2546 external cephalic versions in low-risk pregnant women performed by trained midwifes) where 60% were more than 36 weeks; or in other publications like Ainsworth et al. 2017, with ECV even in 40 weeks of gestation. All of these publications are about ECV complications.

Also, the complication total rate is 17%. This is higher than 6% in the revision published by Kok et al (2008) or Rodgers et al. (2017) about 5% or 2,5% in Neatherlands by Beuckens et al. (2016).

This manuscript can be very interesting and important, but the authors should rewrite their Introduction and Discussion to reference the related literature, because this is not the first publishing data about 38 gw ECVs.

6. PLOS authors have the option to publish the peer review history of their article (what does this mean?). If published, this will include your full peer review and any attached files.

Reviewer #1: No

Reviewer #2: No

---

## [Author Response · Author response to Decision Letter 0]

13 Nov 2020

Journal requirements were double-checked.

'I have read the journal's policy and the authors of this manuscript have the following competing interests:

F.L. is first vice president of the German Society for Gynaecology and Obstetrics (DGGG), council member of the European Board and College of Obstetrics and Gynaecology (EBCOG) and Executive Board Member und Committee chairman of the International Federation of Gynecology and Obstetrics (FIGO).

F.L. and L.J. worked on the first German S3-Guideline for cesarean section published in 2020.

A.Z. declares no relevant conflicts of interest.'

a. Please confirm that this does not alter your adherence to all PLOS ONE policies on sharing data and materials, by including the following statement: "This does not alter our adherence to PLOS ONE policies on sharing data and materials.” (as detailed online in our guide for authors http://journals.plos.org/plosone/s/competing-interests). If there are restrictions on sharing of data and/or materials, please state these.

Please note that we cannot proceed with consideration of your article until this information has been declared.

F.L. is first vice president of the German Society for Gynaecology and Obstetrics (DGGG), council member of the European Board and College of Obstetrics and Gynaecology (EBCOG) and Executive Board Member und Committee chairman of the International Federation of Gynecology and Obstetrics (FIGO).

F.L. and L.J. worked on the first German S3-Guideline for cesarean section published in 2020.

A.Z. declares no relevant conflicts of interest.

This does not alter our adherence to PLOS ONE policies on sharing data and materials.

We included the Update in our cover letter.

3. Review Comments to the Author

Reviewer #1: title - too long

Thank you for your profound comments. They were very helpful in improving our manuscript. 

The title was shortened. The new title is: “External cephalic version in 38 weeks’ gestation in a specialized German single center”

keywords - too many

Less important key words were deleted.

introduction - ECV is mostly performed before 37 GA? 

The wording was infelicitous. Despite recommendations for performing ECV after 37 weeks of gestation, a lot of departments still offer ECV before 37 weeks in Germany. Citations were included in the revised manuscript.

LINE 67 - provide reference for most ecv performed before 37

Citations were included in the revised manuscript.

line 68-70 - is redundant

The passage was shortened, we focused on more present literature.

ECV not offered line 79 - provide reference

Citations were included in the revised manuscript. Kohls et. al. carried out an anonymized online survey asking about preferred primary intervention for ECV in Germany in 2018. Although it seems that ECV is offered nationwide, only for 61% ECV is the preferred primary intervention, for 12.4% of the surveyed hospitals CS is still the preferred intervention.

line 79-81 - more studies to be published??? numerous studies regarding predictors of success. the authors should replace outofdate reference (years 2000-2010) regarding success rate (ezra, ben meir) and replace with more updated reference (31312960, 30941816)

Outdate reference was replaced by updated reference. Only four studies, we think are very important to the topic and were published before 2010 (Hannah et. al 2000, Hutton et. al. 2003, Kainer et. al. 2003, Grootscholten et. al. 2008) were kept in the manuscript.

A prospective analyses? is this trully prospective or retrospective? approval (Ref:176/18) means 2018?

Thank you for pointing out this unclarity. Our study is an interim analysis of an ongoing larger single center study at our department starting from 2004. Our study design is overall prospective, but additional data was collected in a retrospective approach. The additional retrospective data collection was approved by the local ethics committee in 2018. An explanation was added in the revised manuscript.

extremely low success rate of ECV -22% should be discussed and explained

Thank you for addressing this important topic. In the rewritten discussion, we clarified the background of our low success rate. 

reults chapter is written badly - please rewrite.

Thank you for your remarks, the results chapter has been rewritten. We focused on improving the language and cut out the repetitive results. I hope your demands were met.

tables - ages with 2 decimals? what age does 33.13 represents?? parity and gravidity with decimals?

The differences between the two groups in cases of age, parity, gravidity etc. were only visible within the decimals despite being significant. We shortened the decimal digits to only one digit, so the difference is still visible. Without the digits the numbers would be the same, although there is a significant difference.

table 2 looks like it was copied from the SPSS sowtware - please provide a better version of presentation of table

Thank you for the remark. The regression was run using the program Bias. We diminished the reported variables to Odds ratio, standard deviation and Wald’s p.

what does constant mean?

The constant term in regression analysis is the value at which the regression line crosses the y-axis, also known as the y-intercept. It was deleted from the table, as it may cause incomprehension.

vaginal delivery rate of 62% is very low - should be discussed

Thank you for pointing out this unclarity. The vaginal delivery rate of 62% refers to overall vaginal delivery after ECV disregarding the success. To make it clearer, we changed the labeling of the table to: ‘all deliveries after ECV’. 

This number describes cephalic as well as breech vaginal delivery. The rate of cephalic vaginal delivery after successful ECV is 78%, which is higher than the nationwide average on cephalic birth. The rate of vaginal breech birth after unsuccessful ECV exceeds with 60% the nationwide average as well as rates described in previously published data.

line 275-276 claiming novely - what about 31312960?

Thank you for your remark. While Levin et. al. offers ECV as an alternative between CS and breech delivery, at our center we offer a combined approach discussing ECV, vaginal breech delivery and CS to reach a shared decision with the patient.

overall i dont see novelty, maybe unfamiliarity with current literature, therefore a more thorough literature review should be performed in the introduction and discussion and old references (more than 10 years should be discuraged)

The reference list was updated, the introduction and discussion were rewritten. We hope your demands were met in the revised manuscript.

limited new information, low success rate - thus limiting generalizability

We addressed this point more precise in the revised discussion.

Reviewer #2: First of all, this is a very interesting manuscript about management of breech presentation at term. This Care Unit seems to be a very expertise Breech Unit, and provide care about counseling about all possible maneuvers such as ECV, cesarean and vaginal birth. Congratulations for that.

However, it is commented on several occasions that this is the first article published about the safety of ECV in 38 weeks of gestation (lines 39, 49, 85, 275). The study fails to address how the findings relate to previous research in this area. The authors should return to publications such as that of Rodgers et al., 2017 (reference number 32 of the manuscript), where the mean gestational age of the ECV was 37 + 5, or in others such as Beuckens et al., 2015 (an observational study of the success and complications of 2546 external cephalic versions in low-risk pregnant women performed by trained midwifes) where 60% were more than 36 weeks; or in other publications like Ainsworth et al. 2017, with ECV even in 40 weeks of gestation. All of these publications are about ECV complications.

Thank you for your remarks, we followed your recommendation and discussed the articles mentioned above alike your suggestion.

Also, the complication total rate is 17%. This is higher than 6% in the revision published by Kok et al (2008) or Rodgers et al. (2017) about 5% or 2,5% in Neatherlands by Beuckens et al. (2016).

Thank you for addressing this important issue. Our “higher” complication rate compared to previously published data is the result of reporting on all minor bradycardia and CTG-alterations. Beuckens et. al. only reported on bradycardia lasting over 10 minutes, Kok et. al. recommended only to report on bradycardia leading to CS. We discussed the complication rate in the revised paper. 

This manuscript can be very interesting and important, but the authors should rewrite their Introduction and Discussion to reference the related literature, because this is not the first publishing data about 38 gw ECVs.

Thank you for your remarks. We rewrote the Introduction and Discussion as recommended and revised the reference list with more present studies.

---

## [Decision Letter · Decision Letter 1]

28 Mar 2021

PONE-D-20-24091R1

External cephalic version in 38 weeks’ gestation in a specialized German single center

PLOS ONE

Dear Dr. Authors,

Thank you for submitting your manuscript to PLOS ONE. After careful consideration, we feel that it has merit but does not fully meet PLOS ONE’s publication criteria as it currently stands. Therefore, we invite you to submit a revised version of the manuscript that addresses the points raised during the review process.

We look forward to receiving your revised manuscript.

Kind regards,

Salvatore Andrea Mastrolia, M.D.

Academic Editor

PLOS ONE

Journal Requirements:

Reviewers' comments:

Reviewer's Responses to Questions

**Comments to the Author**

1. If the authors have adequately addressed your comments raised in a previous round of review and you feel that this manuscript is now acceptable for publication, you may indicate that here to bypass the “Comments to the Author” section, enter your conflict of interest statement in the “Confidential to Editor” section, and submit your "Accept" recommendation.

Reviewer #3: (No Response)

Reviewer #4: All comments have been addressed

2. Is the manuscript technically sound, and do the data support the conclusions?

Reviewer #3: Yes

Reviewer #4: Yes

3. Has the statistical analysis been performed appropriately and rigorously? 

Reviewer #3: I Don't Know

Reviewer #4: Yes

4. Have the authors made all data underlying the findings in their manuscript fully available?

Reviewer #3: Yes

Reviewer #4: Yes

5. Is the manuscript presented in an intelligible fashion and written in standard English?

Reviewer #3: No

Reviewer #4: Yes

6. Review Comments to the Author

Reviewer #3: 38. Not clear what is meant by breech birth procedures?

39. There has been a paper published this year that demonstrated that ECV at 38 weeks has equal success rates to ECVs performed at 36, 37, 39 and 40 weeks and is equally safe.

48. ECV at 38 weeks.

49. False claim of first evidence.

131. No mention on the technique of ECV used, i.e., forward or backward flip?

154. fetuses.

160. Success rates appear to be skewed by the higher numbers of nulliparous women.

177. Definition of AFI measurements not mentioned, i.e., abundant is 20, 25, 30 cm?

289. The study published in 2021 reported higher success with higher EFW

290-293. Need to be re-written in a clearer way

318. A fetus cannot be referred to as children

327 – 329. No justification/hypothesis written on to why success was higher in larger fetuses.

Reviewer #4: I read with great interest the Manuscript titled “External cephalic version in 38 weeks’ gestation in a specialized German single center ”

The topic of this manuscript falls within the scope of PLOS One.

I was particularly pleased to review this revised version of the manuscript. In my honest opinion, the past reviewers' concerns have been resolved by the Authors. Now its scientific soundness is interesting enough to attract the readers’ attention. The methodology is accurate, and conclusions are supported by the summarized evidence.

However, I believe that the Discussions might benefit from a brief deeping of factors that might or might not improve the ECV successfulness, for example:

- Neuraxial analgesia [ PMID 27131581]

- Tocolysis [PMID 33421816]

- Other prognostic factors [PMID 31369397]

7. PLOS authors have the option to publish the peer review history of their article (what does this mean?). If published, this will include your full peer review and any attached files.

Reviewer #3: No

Reviewer #4: No

---

## [Author Response · Author response to Decision Letter 1]

25 Apr 2021

Reviewer #3: 

Thank you for your profound comments. They were very helpful in improving our manuscript. 

38. Not clear what is meant by breech birth procedures?

Thanks for the remark, the corresponding part has been specified and alternatives for breech birth were described. 

39. There has been a paper published this year that demonstrated that ECV at 38 weeks has equal success rates to ECVs performed at 36, 37, 39 and 40 weeks and is equally safe.

Thank you for pointing out this recent paper. The paper was cited and addressed in the introduction and discussion of the revised manuscript.

48. ECV at 38 weeks.

Thank you very much for your remarks regarding grammar and spelling. We undertook a profound correction. Additionally, the manuscript was proof-read by the suggested English language editing service.

49. False claim of first evidence.

Thanks for this comment. This statement had been adjusted to the current evidence in the revised manuscript. 

131. No mention on the technique of ECV used, i.e., forward or backward flip?

Again, thank you for the helpful remark. The explanation below was added in the revised manuscript.

“The baby was moved upwards with one hand and pushed to perform a forward or backward roll, preferably in the direction with less resistance. In cases of unsuccessful ECV (NECV), both directions were attempted. The direction (backwards or forwards) of successful ECV (SECV) has not been documented.

154. fetuses.

Thanks, we corrected the typo. 

160. Success rates appear to be skewed by the higher numbers of nulliparous women.

Thank you very much for pointing out this important issue. The range of nulliparous women in our population is indeed higher compared to previously published data. We discussed the lower range in the revised manuscript.

177. Definition of AFI measurements not mentioned, i.e., abundant is 20, 25, 30 cm?

As suggested, we explained the cutoff for an abundant AFI score at our center.

289. The study published in 2021 reported higher success with higher EFW.

Thank you for the recommendation. We discussed this recent evidence in our submitted revision.

290-293. Need to be re-written in a clearer way

This part has been extended for more clarity. The whole manuscript has been double-checked for infelicitous wording.

318. A fetus cannot be referred to as children.

Thank you for pointing this out. We corrected this context.

327 – 329. No justification/hypothesis written on to why success was higher in larger fetuses.

Thanks again for the helpful comment. We discussed possible hypothesis for the observed higher success rate in larger fetuses. 

Reviewer #4: 

I read with great interest the Manuscript titled “External cephalic version in 38 weeks’ gestation in a specialized German single center ”

The topic of this manuscript falls within the scope of PLOS One.

I was particularly pleased to review this revised version of the manuscript. In my honest opinion, the past reviewers' concerns have been resolved by the Authors. Now its scientific soundness is interesting enough to attract the readers’ attention. The methodology is accurate, and conclusions are supported by the summarized evidence.

However, I believe that the Discussions might benefit from a brief deeping of factors that might or might not improve the ECV successfulness, for example:

- Neuraxial analgesia [ PMID 27131581]

- Tocolysis [PMID 33421816]

- Other prognostic factors [PMID 31369397]

Thank you very much for your profound comments and remarks. The suggested studies are very interesting and highly relevant. We discussed these articles alike your suggestions. We hope your demands were met in the revised manuscript. The language within the manuscript was also improved with the help of an editing service.

---

## [Editor Report · Decision Letter 2]

21 May 2021

External cephalic version at 38 weeks’ gestation in a specialized German single center

PONE-D-20-24091R2

Dear Authors,

We’re pleased to inform you that your manuscript has been judged scientifically suitable for publication and will be formally accepted for publication once it meets all outstanding technical requirements.

Kind regards,

Salvatore Andrea Mastrolia, M.D.

Academic Editor

PLOS ONE

---

## [Editor Report · Acceptance letter]

11 Aug 2021

PONE-D-20-24091R2 

External cephalic version at 38 weeks’ gestation at a specialized German single center 

Dear Dr. Zielbauer:

I'm pleased to inform you that your manuscript has been deemed suitable for publication in PLOS ONE. Congratulations! Your manuscript is now with our production department. 

Kind regards, 

on behalf of

Dr. Salvatore Andrea Mastrolia 

Academic Editor

PLOS ONE